# Immobilization of Alendronate on Zirconium Phosphate Nanoplatelets

**DOI:** 10.3390/nano13040742

**Published:** 2023-02-15

**Authors:** Anna Donnadio, Geo Paul, Marianna Barbalinardo, Valeria Ambrogi, Gabriele Pettinacci, Tamara Posati, Chiara Bisio, Riccardo Vivani, Morena Nocchetti

**Affiliations:** 1Department of Pharmaceutical Sciences, University of Perugia, Via Del Liceo 1, 06123 Perugia, Italy; 2CEMIN-Centro di Eccellenza Materiali Innovativi Nanostrutturati, University of Perugia, Via Elce di Sotto 8, 06123 Perugia, Italy; 3Department of Sciences and Technological Innovation, University of Piemonte Orientale, “A. Avogadro”, Viale T. Michel 11, 15121 Alessandria, Italy; 4CNR-ISMN, Via P. Gobetti 101, 40129 Bologna, Italy; 5CNR-ISOF, Via P. Gobetti 101, 40129 Bologna, Italy; 6CNR-Istituto di Scienze e Tecnologie Chimiche “Giulio Natta”, Via C. Golgi 19, 20133 Milano, Italy

**Keywords:** zirconium phosphate, gem-bisphosphonate, alendronate, topotactic exchange, solid-state NMR, cytotoxicity

## Abstract

Different amounts of sodium-alendronate (ALN) were loaded into layered zirconium phosphates of alpha and gamma type (αZP and γZP) by means of topotactic exchange reactions of phosphate with ALN. In order to extend the exchange process to the less accessible interlayer regions, ALN solutions were contacted with colloidal dispersions of the layered solids previously exfoliated in single sheets by means of intercalation reaction of propylamine (for αZP) or acetone (for γZP). The ALN loading degree was determined by liquid P-nuclear magnetic resonance (NMR) and inductively coupled plasma (ICP), and it was reported as ALN/Zr molar ratios (Rs). The maximum R obtained for γZP was 0.34, while αZP was able to load a higher amount of ALN, reaching Rs equal to 1. The synthesized compounds were characterized by X-ray powder diffractometry, scanning electron microscopy (SEM), solid-state NMR, and infrared spectroscopy. The way the grafted organo-phosphonate groups were bonded to the layers of the host structure was suggested. The effect of ZP derivatives was assessed on cell proliferation, and the results showed that after 7 days of incubation, none of the samples showed a decrease in cell proliferation.

## 1. Introduction

The research on metal phosphonates has grown considerably over the last few decades [1] due to an increased demand for new functional and versatile solids for applications in many fields of materials chemistry. Specifically, the design of organically modified surfaces allows access to materials possessing tunable properties. This modification can be performed by grafting onto the surface appropriate functional molecules, thus, providing better control of the density and orientation of the organic component at the surface [2].

As far as the synthetic approaches are concerned, zirconium phosphonates (ZPs) can be prepared according to two main procedures: direct synthesis and a topotactic exchange reaction that leaves the layer structure essentially unchanged. There are many examples in the literature on the topotactic exchange reactions involving γ-type zirconium phosphate (Zr(PO_4_)(H_2_PO_4_)·2H_2_O, hereafter γZP [3], while few papers report the topotactic synthesis of α-type zirconium phosphonates from α-zirconium phosphate (Zr(HPO_4_)_2_·H_2_O, hereafter αZP, by using both severe conditions of temperature and pressure and molten phosphonic acid [4] and, more recently, quicker and milder conditions [5]. The great variety of functional groups based on both monophosphonates and diphosphonates that can be incorporated into these metal phosphonate structures leads to unique chemistry [6]. However, until now, no example of zirconium phosphonates based on geminal phosphonates has been reported. The gem-bisphosphonate (BP) family constitutes a class of drugs used for the treatment of bone diseases by promoting bone mineralization and inhibiting bone resorption. They are characterized by a P-C-P bond, where Ps are phosphonate (-PO_3_^=^) groups, and consequently, they are analogs of pyrophosphate but are resistant to chemical hydrolysis. Individual BPs differ in the two covalently bound side chains, R1 and R2, which complete the tetra valence of the carbon atom [7]. Amino BPs, such as alendronate (ALN) (Figure 1), can be delivered by systemic delivery or otherwise by local delivery. Although the adsorption of BPs on the implant is a simple procedure, it suffers from the out-of-control release of BPs over the long term and may influence specific physiological responses [8].

The effective immobilization of bioactive species on solid substrates without changing their original structure and bioactivity, which allows for maintaining drug release over a sustained period, has attracted increasing interest among researchers along with rapid advances in biodevices [9,10]. The topotactic exchange method can be considered an alternative to the preparation, via the intercalation reaction, of drug delivery systems based on ZP compounds. In this scenario, the intercalation of drugs such as doxorubicin [11] and 5-fluorouracil [12], chlorhexidine [13], methylene blue as a photosensitizer [14], and NorA efflux pump inhibitors [15] was successfully performed on ZP supports. Thus, ZPs have shown promising results as nanocarriers for drug delivery and other biotechnological applications [1]. This research highlights the potential of the topotactic exchange method to prepare modified ZP in which ALN is covalently anchored on the surface of the supports. In this way, it is possible to obtain an efficient load of the drug and a mediated release, reducing, at the same, the risk of the out-of-control release of BPs.

Here, novel zirconium phosphate phosphonates prepared by a topotactic exchange reaction of αZP and γZP with ALN are reported. The loading of ALN in αZP and γZP is determined by elemental analysis and liquid NMR. The connectivity of ALN to zirconium in the α and γ layer is investigated, in detail, by X-ray diffraction and solid-state NMR. Finally, the biocompatibility of αZP and γZP with ALN is investigated.

## 2. Materials and Methods

### 2.1. Materials

Alendronate sodium trihydrate C_4_H_12_NaNO_7_P_2_·3H_2_O was purchased from Farmalabor (Canosa di Puglia (BT), Italy). All other reagents were purchased from Merck (Milano, Italy) and were used as received without further purification.

### 2.2. Synthesis of αZP and γZP

Semicrystalline αZP was synthesized as reported in [16]. Typically, a weighed amount of zirconium oxide chloride was dissolved, under stirring at ambient conditions, in 35 mL of an aqueous solution of oxalic acid, where Zr(IV) concentration was 0.1 M and the H_2_C_2_O_4_/ Zr molar ratio was 10. Then, a volume of 14.8 M H_3_PO_4_ was added so that the P/Zr molar ratio was 6. The solution obtained was heated for 24 h at 80 °C in a closed bottle. The resulting precipitate was separated from the solution by centrifugation at 3000 rpm, washed three times with a diluted HCl solution, dried overnight at 80 °C, and, finally, stored in a desiccator at 53% relative humidity (RH). Crystalline γZP was prepared as reported in [17]. Briefly, 50 mL of an aqueous solution of 0.034 M ZrOCl_2_·8H_2_O and a concentrated HF solution (F/Zr molar ratio = 6) were mixed with a 50 mL solution of 0.94 M NH_4_H_2_PO_4_·H_2_O so that the molar ratio P/Zr was 55. The solution was placed at 80 °C in a closed bottle for 5 days. The precipitate obtained was separated from the solution by centrifugation at 3000 rpm, washed three times with 10^−3^ M HCl, dried overnight at 80 °C, and, finally, stored in a desiccator over a saturated BaCl_2_·2H_2_O solution (90% RH).

### 2.3. Delamination of αZP and γZP

αZP was delaminated following the procedure reported in [18]. Briefly, 1 g of αZP was suspended in 100 mL of water, and then, 33.2 mL of 0.1 M *n*-propylamine (N/P molar ratio of 1:1) was added under vigorous magnetic stirring. A volume of 10 mL of HCl 1M was added to the solution (so as to reach pH < 2) to remove the base and regenerate the hydrogen form of ZP. The resulting solid was separated by centrifugation at 5000 rpm from the solution and washed with copious amounts of water until the samples were free of chloride ions. This gel sample labeled as αZPg was stored in a closed container. To determine the content of solid in the gel, a weighted amount of the gel was dried in an oven at 100 °C up to constant weight. The content of the solid was ca. 7 wt %. As for γZP, the colloidal dispersion was obtained following the procedure reported in the literature [12]: 0.5 g of solid was suspended in 70 mL of water/acetone (1:1 *v*/*v*) at room temperature in a closed container and then kept at 80 °C for 2 h under magnetic stirring [17]. The dispersion was labeled as γZPg.

### 2.4. Preparation of ZP ALN-Based Compounds

A weighted amount of αZPg or γZPg containing 1 g of solid was added to a 0.1 M aqueous solution of ALN using different ALN/Zr molar ratios (Rs), ranging from 0.5 to 2. The mixture was kept at 80 °C for five days in a closed vessel. The solid was recovered and washed several times with deionized water to remove any remaining unreacted ALN and, finally, dried at 50 °C overnight. All samples were stored over a saturated solution of BaCl_2_·2H_2_O. The solids were labeled as αZPR and γZPR, where R = 0.5, 1, 2.

### 2.5. Analytical Procedures

ICP was used to determine zirconium and phosphorus contents using a Varian Liberty Series II instrument working in axial geometry (Varian 700-ES series, Santa Clara, CA, USA). The samples were mineralized with some drops of concentrated hydrofluoric acid and concentrated nitric acid.

Carbon, nitrogen, and hydrogen contents were obtained by elemental analysis using an EA 1108 CHN, (Fisons instrument, Glasgow, UK).

X-ray powder diffraction (XRD) patterns were collected with the Cu-Kα radiation on a Bruker D8 Advance diffractometer (Bruker AXS GmbH, Karlsruhe, Germany) equipped with a Lynxeye XE-T detector. The long fine focus (LFF) tube was operated at 40 kV and 40 mA. The samples were carefully side-loaded onto a zero-background sample holder to minimize preferential orientations of the microcrystals. The phase identification was performed using Bruker DIFFRAC.EVA V5 software equipped with the COD database (software version 2.0 up, © 2010–2019 Bruker AXS GmbH, Karlsruhe, Germany).

Field-emission scanning electron microscopy (FE-SEM) was employed to collect images using an LEO 1525 ZEISS instrument (Jena, Germany) working with an acceleration voltage of 15 kV. The elemental mapping of metals in samples was conducted by using energy-dispersive X-ray spectroscopy (EDX) (Ewing, NJ, USA).

Attenuated total reflection (ATR) FT-IR measurements were carried out using a Shimadzu IR-8000 spectrophotometer (Kyoto, Japan). The spectral range measured was 400 to 4000 cm^−1^, with a spectral resolution of 4 cm^−1^ acquiring 100 scans.

Solid-state NMR spectra were collected on a Bruker Avance III 500 spectrometer (Faellanden, Switzerland) and a wide bore 11.75 Tesla magnet with operational frequencies for ^1^H, ^13^C, and ^31^P of 500.13, 125.77, and 202.46 MHz, respectively. A 4 mm triple resonance probe in double resonance mode with magic angle spinning (MAS) was used in all the experiments. The samples were loaded in a zirconia rotor, closed with a Kel-F cap, and spun at a MAS rate of 15 kHz. ^31^P MAS spectra were collected with a 90-degree pulse, and the magnitude of the radio frequency field was 70 kHz with ^1^H decoupling during acquisition. Spectra were recorded with a spectral width of 100 kHz, and 16 transients were accumulated at 300 K using a relaxation delay between accumulations of 300 s.

For the ^13^C{^1^H} cross-polarization (CP) MAS experiments, proton radio frequency (RF) fields of 62 and 33 kHz were employed for initial excitation and decoupling, respectively. During the CP period, the ^1^H RF field was ramped using 100 increments, whereas the ^13^C RF field was maintained at a constant level. A moderate ramped RF field of 55 kHz was used for spin locking, while the carbon RF field (40 kHz) was matched to obtain the optimal signal. During the acquisition, the protons were decoupled from the carbons by using a Spinal-64 decoupling scheme. A CP contact time of 2 ms and a delay between scans of 10 s (for ALN) and 1 s were used. Spectra were acquired with a spectral width of 42 kHz, and between 256 (for ALN) and 5120 transients were accumulated at 300 K. All chemical shifts were reported using δ scale and were externally referenced to ammonium dihydrogen phosphate at 0.8 ppm for ^31^P and TMS at 0 ppm for ^13^C.

### 2.6. Cell Cultures

MG-63 (human bone osteosarcoma) (Sigma Aldrich, Milano, Italy) was cultured in minimum essential medium with 10 vol % fetal bovine serum, 2 mM L-glutamine, 0.1 mM MEM nonessential amino acids (NEAAs), 100 U·mL^−1^ penicillin, and 100 U·mL^−1^ streptomycin in an incubator set at 37 °C with 5% CO_2_. Cells were seeded on samples at a density of 5 × 10^4^ cells·cm^2^ [19,20].

### 2.7. Resazurin Reduction Assay

The resazurin reduction assay assesses cell viability; the resazurin solution was inlaid directly to the samples with 10 vol % and incubated for 4 h. All the details are described by Barbalinardo et al. [21] The plate reader used was a Thermo Scientific Varioskan Flash Multimode Reader (Waltham, MA, USA).

## 3. Results and Discussion

Gamma zirconium phosphate has a layered structure, and each layer is made of two planes of zirconium coordinated by tetradentate trivalent PO_4_ groups between them and bidentate monovalent H_2_PO_4_ groups on the external part of each layer [22]. As a result, these external dihydrogen phosphate groups can be easily replaced by other phosphonate groups by contacting γZP with a solution of the proper incoming phosphonic acid through a topotactic reaction, which leaves unaltered the remaining (covalent) connectivity of gamma layer [23]. The topotactic process is especially efficient when colloidal dispersions of exfoliated γZP are used as a substrate, where the solution has free access to the surface of all layers. The amount of replacement is gradual with the formation of solids with the general formula ZrPO_4_(H_2_PO_4_)_1−*x*_(O_2_PJJ’)*_x_*, in which J and J’ can be H, OH, or organic groups. The *x* value generally depends on the steric and electronic features of the incoming group and, given the above formula, at maximum can be 1.

Zirconium phosphate-ALN, with an increasing content of ALN, was prepared using topotactic exchange reactions by contacting exfoliated γZP with solutions containing different relative amounts of alendronate (R values of 0.5, 1, and 2). The samples prepared with R ≥ 1 showed the same XRD profile and composition, determined by liquid NMR and ICP measurements, suggesting that γZP1 contained the maximum amount of ALN supported by the gamma structure. No further phosphate groups, with respect to γZP1, could be exchanged by using R = 2. Therefore, only γZP05 and γZP1 samples underwent further analysis.

The XRD patterns of γZP-ALN samples in comparison with that of γZP are shown in Figure 1. The topotactic exchange of phosphate groups by ALN caused some modifications in the diffraction profile of the samples with respect to the pristine γZP: an increased broadening of the reflections, the shift of the first reflection towards lower 2theta angles, and the presence of more than one phase. The peak broadening can be associated with the loss of order inside the structure; the reaggregation of lamellae functionalized with ALN probably caused a disordered re-staking of the layers due to the bulky structure of the phosphonic acid and to its random occupation of the surface sites of γ-sheets. As expected, given the dimension of ALN, the interlayer distance of the samples increased from 12.2 Å of pure γZP to 15.4 Å. The additional reflections at 7.7° 2θ (d = 11.4 Å) may likely be ascribable to the half-sodium-exchanged forms of γZP, as previously reported [24]. The formation of this phase can be explained considering that some phosphate groups remained not exchanged, and the sodium cations, present as counter anions in ALN (Figure 1), exchanged the most acid protons of phosphate groups. The reflection at about 11.4 Å disappeared (Appendix A) after washing the samples with 0.1 M HCl as a result of sodium exchange with the proton; this confirms the attribution of this reflection to the hydrated half-sodium-exchanged forms of γZP.

αZP is also a layered compound, and the structure of the layers is simpler than γZP, as each α-layer consists of a plane of zirconium atoms coordinated by tridentate HPO_4_ groups, which occupy both faces of the layers. Although these hydrogen phosphate groups are tightly bonded, their topotactic replacement with other phosphonate groups is still possible, especially in colloidal dispersions of exfoliated αZP samples. The samples of αZP exchanged with ALN were also prepared from the colloidal dispersion of αZP by using R values of 0.5, 1, and 2. As can be seen from the patterns in Figure 1, all hybrid compounds based on αZP have a fairly disordered structure.

The low-angle region, which can give us information on the interlayer distance between the different phases, can be described by three broad reflections approximately at d = 12.6, 11.1, and 8.4 Å, plus one peak at 7.6 Å, which is only present in αZP05 and corresponds to a fraction of unreacted αZP.

At higher R values, there are no reflections typical of the starting αZP, except for the peak at 33.8° 2θ, which is the characteristic 020 reflection of αZP-type structures, because it is associated with Zr-Zr atom separation inside the layers. It must be highlighted that this peak at 2θ = 33.8° (d = 2.7 Å) is present in all XRD patterns. This may indicate that the exchange of monohydrogen phosphate with the phosphonate groups of ALN occurred, in any case, without altering the framework of the α-layer [18]. When the samples were washed with 0.1 M HCl, two very broad reflections at d ≈ 11.7 Å and 8.2 Å appeared (Appendix A) in all samples, while in the αZP05 pattern, the peak characteristic of the pristine structure was still present.

Although the low degree of crystallinity of the powders does not allow the resolution of the structure, a structural arrangement can be assumed based on the models built, taking into account the interlayer distances observed. As for gamma-modified zirconium phosphate, a plausible arrangement is proposed in Figure 2a. Because the interatomic distance between the two phosphorus atoms within the ALN moiety is shorter than that of the two adjacent phosphate groups in the gamma layers, in this model, we assume that the ALN molecules are linked only by one phosphonate to the layer and are arranged so that the free phosphonate groups of the ALN point toward the interlayer region. The interlayer distance determined based on this arrangement nearly matches with that found in the X-ray diffraction analysis.

The extent of the topotactic exchange, in terms of the ratio between the phosphate and the phosphonate groups present in the solids, was determined by evaluating the ratio of the areas of the observed peaks in the ^31^P liquid-state NMR spectra. Two main resonances can be observed at approximately δ = 18–19 ppm and one at −0.5 ppm, which are characteristic of phosphonate and phosphate groups, respectively [25,26]. The P:Zr molar ratio was determined by ICP measurements and was combined with NMR data in order to obtain the ALN/Zr molar ratios in the solids (Rs) reported in Table 1.

As for gamma-type materials, Rs reaches 0.34 regardless of the amount of ALN in the starting mixture. While in the alpha-type materials, increasing the amount of ALN in the reaction mixture, Rs increases from 0.48 to 0.86 to 1 for R = 0.5, 1, and 2, respectively.

The ZP-ALN samples were examined by FT-IR, and the spectra were compared with those of pristine ZP (Figure 3). Regarding the γ-compounds, all samples showed spectra dominated by strong absorption at 950–1100 cm^−1^ characteristic of the P-C and P-O stretching modes [27]. In addition, in the γZP-ALN spectra, there are two weak bands at 1500 cm^−1^ related to the asymmetric bending of the NH_3_^+^ group and at 3000–3100 cm^−1^ due to the stretching of the N-H bond [28,29]. These bands prove the presence of the -NH_3_^+^ groups arising from the protonation of the terminal NH_2_ groups of ALN by the phosphate or phosphonate groups. Finally, at about 2930 cm^−1^, a fairly wide band characteristic of the C-H stretching of the ALN alkyl chain can be observed.

The FT-IR spectra of the exchanged ZPs, together with the parent αZP, are shown in Figure 3. As can be seen, again, strong absorptions at 950–1100 cm^−1^, characteristic of the P-C and P-O stretching modes, are present in all samples [30]. The substantial red shift of the P-O frequency is credible because the binding of ALN to the ZP layer could significantly decrease the frequency of the P-O stretching mode [31]. Moreover, in the spectra of the composites, the presence of -NH_3_^+^ groups, due to the protonation of the terminal NH_2_ groups of the ALN by the phosphate or phosphonate groups, are provided by two weak bands at 1500 cm^−1^ and 3000–3100 cm^−1^. In particular, we found that the first band is related to the asymmetric bending and the second to the stretching of the N-H bond, which is in agreement with the data in the literature, as reported above. The deprotonation of the monohydrogen phosphate groups can also be confirmed by the shift in the absorption at 1050 cm^−1^ of the ZP (Figure 3B) (characteristic of HPO_4_^2−^) to 980 cm^−1^ (characteristic of PO_4_^3−^), whose intensity increases by increasing the amount of NH_2_ groups [32]. Finally, at about 2930 cm^−1^, a fairly broad band characteristic of the C-H stretching of the alkyl chain of ALN can be seen.

Scanning electron micrographs (SEMs) reveal that the composites based on γZP formed thin micron-sized platelets similar to pristine γZP [33]. The morphology was rectangular with a thickness of some tens of nm (Figure 4). No evident changes in the morphology or size of the platelets could be observed after the topotactic exchange, even if they were a little more irregular in shape.

As for the composites based on αZP, the particles are similar in size and morphology to the pristine ZP. Again, the morphology was retained after ion exchange with ALN groups to yield hybrid αZP. The images showed small, irregularly shaped platelets with a size lower than 1 μm (Figure 5).

In both series, the EDX analysis shows a homogeneous distribution of nitrogen, carbon, phosphorus, and zirconium elements, thus, attesting that the topotactic exchange occurred successfully in both cases. The signal of the sodium element was also detected by EDX, confirming that it is included in the structure to some extent (Appendix A).

^31^P NMR spectroscopy is an excellent tool for investigating the local coordination states of zirconium phosphate phosphonates [34]. The gamma-type samples were characterized by ^31^P MAS NMR spectroscopy, and the obtained results are shown in Figure 6.

The ^31^P MAS NMR spectrum of γZP (Figure 6a) showed two intense resonances at δ_P_ = −27.5 ppm and −9.8 ppm due to the P(ZrO)_4_ and P(ZrO)_2_(OH)_2_ groups, respectively, typical of this compound, as reported by Clayden [35]. Moreover, weak resonances were also detected at δ_P_ = −16.3 ppm, −15.0 ppm, and −13.3 ppm due to defect [36]. The relative proportion of the population distribution of phosphate to dihydrogen phosphate species estimated from the quantitative ^31^P MAS NMR data agrees with the theoretical value. Therefore, the defect sites are associated with dihydrogen phosphate species. The ^31^P MAS NMR spectrum (Figure 6b) of ALN-Na exhibited two resonances at δ_P_ = 17.6 ppm and 22.2 ppm, which are characteristic of the two crystallographically distinct phosphonate groups present in the sample.

The ^31^P MAS NMR spectra of γZP05 and γZP1 (Figure 6c,d) show two sets of resonances in the range of −30 to −10 ppm and −5 to 25 ppm, respectively, due to phosphate and phosphonate units. The chemical shifts associated with P(ZrO)_4_ and P(ZrO)_2_(OH)_2_ groups in these samples are slightly different from that of the parent γZP sample. These differences are related to the changes in the next nearest neighbor environments of ^31^P. On the other hand, resonances due to phosphonate motifs originating from ALN that are formed due to the topotactic replacement of the P(OZr)_2_(OH)_2_ units are broader due to the chemical shift distribution associated with diverse ^31^P species. Various phosphonate motifs are assigned in Figure 6 according to their ^31^P chemical shifts. However, the extraction of quantitative distribution is rendered difficult. Furthermore, the relative proportion of the population distribution of phosphorus species was extracted from the quantitative ^31^P MAS NMR data, and the estimated phosphate-to-phosphonate ratios are 1/(0.30 ± 0.1) and 1/(0.50 ± 0.1) for γZP05 and γZP1, respectively. A greater extent of the topotactic exchange is noted in γZP1 at the expense of the P(OZr)_2_(OH)_2_ groups. The phosphate/phosphonate ratios are in good agreement with those found by ^31^P liquid-state NMR (see Table 1).

Subsequently, the stability of the ALN during the topotactic exchange reactions was monitored using ^13^C CPMAS NMR spectroscopy (Figure 7).

Compared to ALN ^13^C resonances, the peaks in γZP05 and γZP1 are broadened due to the chemical shift distribution arising from the structural heterogeneities. However, the integrity and stability of organophosphonate units belonging to ALN are retained after the topotactic exchange reactions, as confirmed by the NMR measurements. In addition, the ^23^Na MAS NMR data reveal that a significant amount of ^23^Na is retained in both the samples, γZP05 and γZP1, and is expected to influence the degree of protonation of phosphates and phosphonates.

The ^31^P MAS NMR spectra of different compounds are reported in Figure 8.

^31^P MAS NMR spectrum for αZP consists of a main isotropic resonance centered at −19.1 ppm, with a weak shoulder at −17.3 ppm [37].

Although αZP has two crystallographically inequivalent monohydrogen phosphate anions, only the sharp peak at −19.1 ppm can be assigned to O_3_POH groups [38]. Therefore, the origin of the weak shoulder could be due to the different hydration states around O_3_POH [39].

The ^31^P NMR spectra for αZP05 (Figure 8b) indicate the existence of two sets of resonances due to phosphate and phosphonate units. Among them, the sharp peak at −19.1 ppm is due to the parent αZP. The peak at −21.8 ppm can be assigned to the phosphate units present in the new phosphate-ALN zirconium phase formed [40]. Moreover, the ^31^P MAS NMR spectrum shows two broad resonances at around δ_P_ = 5.8 ppm and 11.5 ppm assigned to phosphonates due to PC(ZrO)_2_OH and PC(ZrO)(OH)_2_ units, respectively. When the R value increased to 1 (Figure 8c) and 2 (Figure 8d), the resonances associated with these phosphonate motifs increased in intensity significantly. In addition, the weak shoulder at approximately 2 ppm indicated these samples also contained a significant amount of PC(ZrO)_3_ groups. The estimated phosphate-to-phosphonate ratios are 1/(0.65 ± 0.1), 1/(2.0 ± 0.1), and 1/(3.5 ± 0.1) for αZP05, αZP1, and αZP2, respectively, implying a very high level of topotactic exchange in αZP2. These values are, also in this case, in agreement with those obtained by ^31^P liquid-state NMR (see Table 1).

Successively, the stability of the ALN during the topotactic exchange reactions was confirmed by employing ^13^C CPMAS NMR spectroscopy (Figure 9). It was noted that the decomposition of ^13^C resonance due to C3 carbon in topotactic exchanged samples, especially in αZP2, displayed at least two components. Such distinguished chemical shifts provide unambiguous information on the local order as well as symmetry, and the C3 carbon might be located in two different crystallographic sites. Furthermore, the ^23^Na MAS NMR data reveal that only traces of ^23^Na are retained in αZP05, αZP1, and αZP2, confirming the successful topotactic exchange reactions.

To obtain local structural conformation information and molecular packing motifs, a 2D ^31^P-^1^H FSLG heteronuclear correlation (HETCOR) NMR experiment was performed on αZP2. This experiment correlates ^1^H–^31^P spin pairs in close spatial proximity [41,42]. The 2D ^31^P-^1^H HETCOR spectrum of αZP2 obtained with a relatively short cross-polarization contact time of 1 ms allowed the proton chemical shifts in all the phosphonate hydrogens (from ALN-Na) to be determined (Figure 10A). In addition, protons belonging to the O_3_POH units of the residual αZP are visible at around 8.2 ppm (red box). More importantly, a strong correlation peak for phosphate units (from the new phosphate-ALN zirconium phase) with protons at 7.1 ppm can be observed. This correlation peak is also visible on the HETCOR spectrum recorded with a very short contact time of 0.05 ms (data not shown).

These phosphate units (at −21.5 ppm) also show HETCOR signals with entire ALN-Na protons as well as with water and hydroxyl protons (Figure 10A). The slices extracted from the 2D spectrum (Figure 10B) reveal the direct proton correlations associated with phosphate and phosphonate species. This observation is clear proof of the close proximity between phosphate and phosphonate groups and, thus, the hybrid nature of the phosphate-ALN zirconium phase. Moreover, the distinct ^1^H resonance intensity profiles associated with these two ^31^P sites reveal their proton neighborhoods. The ^31^P of phosphonate units shows a strong correlation with C2 and C3 methylene protons as well as with hydroxy/water protons, while the ^31^P of phosphates makes a strong correlation with NH_2_/NH_3_ protons as well as C4 methylene protons. Because such cross-peaks were also observed with the short 0.05 ms contact time HETCOR experiment, this reflects closely bound ^31^P–^1^H spin pairs located at the inter-layer proximity. Furthermore, the detection of two ^1^H peaks associated with NH_2_ and NH_3_ groups suggests the partial protonation of the ALN chains, which, therefore, provides additional evidence for their close proximity to acidic protons, probably from the phosphate units. Thus, the 2D HETCOR NMR data provide strong experimental evidence that ALN chains adopt an interlayer gallery projected structure with a partial donor-to-acceptor interlayer stacking arrangement, which locks them in space closer to phosphate groups on the other side of the inorganic ZP layer.

The effect of ZP on cell proliferation was assessed by resazurin reduction assay on MG63 cells. As shown in Figure 11, after two days of incubation, the cells in the presence of both αZP and γZP show a viability of more than 75%. After 7 days of incubation, none of the samples showed a decrease in cell proliferation, but in the three samples, αZP, αZP0.5, and γZP1 showed higher viability than the control. It seems that αZP and γZP themselves have no cytotoxic effect on these cells and can be used for bone regeneration. It is interesting to note that after 7 days in αZP, the presence of ALN leads to a slight decrease in cell viability; however, in γZP samples with increasing concentrations of ALN, we note a slight increase in osteoblast proliferation. Further studies on different cellular lines, apoptosis by live/death assay, and mitochondrial activity by MTT assay [14] will be essential to confirm these observations. Moreover, to understand possible cell structure changes, the cell morphology will be investigated by means of confocal microscopy and atomic force microscopy measurements [43].

## 4. Conclusions

The modification of αZP and γZP were successfully performed with alendronate by a topotactic reaction starting from their colloidal dispersion. This was the first time that the preparation of such types of organo-modified zirconium phosphonates and their characterization were reported. In the case of αZP, the portion of exchanged phosphonate groups could be controlled by the ratio of starting reagents in the reaction mixture, while in the case of gamma-type zirconium phosphate, the amount of alendronate remained nearly the same regardless of the ALN:Zr molar ratio of the starting mixture. The basal spacing of the gamma-based products was roughly the same in all cases, also showing a fraction of the sodium form of γZP, while for alpha compounds, a low amount of alendronate allowed for reaching only a partial exchange, leading to obtaining a mixture of exchanged and unchanged semicrystalline zirconium phosphate. The solid-state NMR analysis indicates that no residual signals from crystalline ALN were visible in all ALN-derivatives. In these samples, new broad resonances with different chemical shifts appeared due to phosphonates differently bonded to the gamma layers. In particular, phosphonate coordination to Zr layers was dominated by mono-coordination and, to some extent, bis-coordination. As for the αZP-ALN derivatives, the pristine ZP was still present when a low amount of ALN was used, while the disappearance of the unexchanged αZP phase in αZP1 and αZP2 was noted. Moreover, a high phosphonate exchange degree was proven, demonstrating that the α-based samples have a high capacity to bond with the organic moiety. The aggregation of the alpha and gamma ZP nanoparticles, due to their different sizes, gave rise to different surface roughness. However, the cell proliferation data on both ZPs were not sufficient to assess that the minimal differences observed in the cell viability were due to the differences in the surface roughness of the supports. The contribution of the specific surface and hydrophilicity will be considered.

These merits combined with the very low toxicity make ZP-ALN composites highly promising for therapeutic applications, such as ALN delivery vehicles, and, therefore, could be easily exploited in future studies to enhance their performances in innovative scaffolds.

## Data Availability

The data presented in this study are available on request from the corresponding author.

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
