# Peer review of "Immobilization of Alendronate on Zirconium Phosphate Nanoplatelets"

_nanomaterials, 2023, doi:10.3390/nano13040742_

Round 1

Reviewer 1 Report

The authors provide a very detailed piece of work on the synthesis of zirconoum phosphate and sodium alendronate composite and a very complete characterization.  The synthesis and characterization part of the surfaces is very complete using a very appropriate combination of structural, chemical and surface sensitive techniques, which makes the work very strong. The part on cell proliferation and viability is a bit weaker with only a resazurin test included. At the end of the paper, the authors should include potential techniques to probe cell adhesion and proliferation on the their synthesized surfaces with their respective references. For instance, the authors should check the following references: 

Photodynamic effect of Zirconium phosphate biocompatible nano-bilayers containing methylene blue on cancer and normal cells

doi: 10.1038/s41598-019-51359-7

QCM-D Study of Time-Resolved Cell Adhesion and Detachment: Effect of Surface Free Energy on Eukaryotes and Prokaryotes

doi.: 10.1021/acsami.0c00353

In Figures 4 and 5 differences in surface morphology in alpha and gamma ZPs are evident, in particular, the presence of porosity for the former ones. The authors should comment on how surface topography affects first of all cell adhesion, spreading and proliferation and explain how they can tackle this question.

Author Response

Dear Editor,

Thank you for giving us the opportunity to submit a revised draft of the manuscript “Immobilization of Alendronate on Zirconium Phosphate Nanoplatelets” for publication in Nanomaterials. We appreciate the time and effort that you and the reviewers dedicated to providing feedback on our manuscript and are grateful for the insightful comments on and valuable improvements to our paper. We have incorporated most of the suggestions made by the reviewers. The changes suggested by reviewers are highlighted within the manuscript. Please see below, in italics, for a point-by-point response to the reviewers’ comments and concerns. All page numbers refer to the revised manuscript file with tracked changes.

Reviewer 1:

The authors provide a very detailed piece of work on the synthesis of zirconoum phosphate and sodium alendronate composite and a very complete characterization.  The synthesis and characterization part of the surfaces is very complete using a very appropriate combination of structural, chemical and surface sensitive techniques, which makes the work very strong. The part on cell proliferation and viability is a bit weaker with only a resazurin test included. At the end of the paper, the authors should include potential techniques to probe cell adhesion and proliferation on the their synthesized surfaces with their respective references. For instance, the authors should check the following references: 

1)Photodynamic effect of Zirconium phosphate biocompatible nano-bilayers containing methylene blue on cancer and normal cells doi: 10.1038/s41598-019-51359-7

2) QCM-D Study of Time-Resolved Cell Adhesion and Detachment: Effect of Surface Free Energy on Eukaryotes and Prokaryotes doi.: 10.1021/acsami.0c00353

We thank the referee for all the comments. Our idea in this work is to present the synthesis and the characterization of novel zirconium bis-phosphonates. In order to investigate the possible use of these compounds in biomedical applications the Cytotoxicity of the pristine and modified alphaZP and gammaZP was evaluated. We agree with the referee that we still have to investigate the typology of death, mitochondrial activity and cellular morphology. So, we went on to modify the text taking a cue from the suggested literature: pag 13-14, line 462-465.

“Further studies on different line cellular and also apoptosis by live/death assay, mitochondrial activity by MTT assay will be essential to confirm these observations. Moreover, to understand possible cell structure changes.the cell morphology will be investigated by means of confocal microscopy and atomic force microscopy measurements.

 In Figures 4 and 5 differences in surface morphology in alpha and gamma ZPs are evident, in particular, the presence of porosity for the former ones. The authors should comment on how surface topography affects first of all cell adhesion, spreading and proliferation and explain how they can tackle this question.

Cell adhesion on matrices depends on the surface topography such as the surface roughness, specific surface and hydrophilicity. As mentioned by the Reviewer, the aggregation of the alpha and gamma ZP nanoparticles, due their different size, gives rise to a different surface roughness. However, the cell proliferation data on both ZPs, are not sufficient to assess that the minimal differences observed in the cell viability are due to the differences in surface roughness of the supports. The contribution of the specific surface and hydrophilicity should be also considered. This consideration was added in the text at page 14 line 490-495.

Reviewer 2 Report

The manuscript nanomaterials-2201041 entitled "Immobilization of Alendronate on Zirconium Phosphate Nanoplatelets" is submitted by Anna Donnadio, Geo Paul, Marianna Barbalinardo, Valeria Ambrogi, Gabriele Pettinacci, Tamara Posati, Chiara Bisio, Riccardo Vivani, and Morena Nocchetti for publication in the journal Nanomaterials-MDPI.

The research deals with the synthesis and characterization of layered zirconium phosphates loaded with sodium-alendronate. The materials are synthesized by topotactic reactions from colloidal dispersions and characterized by ICP, elemental analysis, XRD, SEM, FT-IR, and solid-state NMR. Finally, cell viability and cytotoxicity tests are carried out in contact with MG-63 cells, which are human bone osteosarcoma cells.

The synthesis process and the results are interesting and well described. The characterizations provide a good description of the chemical structure of this innovative material.

In my opinion, this article could be worthy of publication in the journal Nanomaterials – MDPI after considering the following minor points:

 - The acronyms must be defined the first time they appear in the text (e.g., NMR, SEM…).

 - "Powder XRD" or "PXRD" is better than "XRPD", and more usual for the readership.

 - In line 242, please add the word "diffraction" to describe "X-ray diffraction analysis".

 - Energy dispersive X-ray spectroscopy is mentioned as EDS or EDX in the text. Please homogenize it, using only one acronym.

  - more comparison with the literature on similar topics is necessary. Why are the properties of this innovative material valuable and more efficient than the others? For which applications? For example, there is no discussion (and no references) in section 3.1, but only a description of some results. Why are these biological results interesting for the study? Please clarify it.

Author Response

Dear Editor,

Thank you for giving us the opportunity to submit a revised draft of the manuscript “Immobilization of Alendronate on Zirconium Phosphate Nanoplatelets” for publication in Nanomaterials. We appreciate the time and effort that you and the reviewers dedicated to providing feedback on our manuscript and are grateful for the insightful comments on and valuable improvements to our paper. We have incorporated most of the suggestions made by the reviewers. The changes suggested by reviewers are highlighted within the manuscript. Please see below, in italics, for a point-by-point response to the reviewers’ comments and concerns. All page numbers refer to the revised manuscript file with tracked changes.

Reviewer 2:

The manuscript nanomaterials-2201041 entitled "Immobilization of Alendronate on Zirconium Phosphate Nanoplatelets" is submitted by Anna Donnadio, Geo Paul, Marianna Barbalinardo, Valeria Ambrogi, Gabriele Pettinacci, Tamara Posati, Chiara Bisio, Riccardo Vivani, and Morena Nocchetti for publication in the journal Nanomaterials-MDPI.

The research deals with the synthesis and characterization of layered zirconium phosphates loaded with sodium-alendronate. The materials are synthesized by topotactic reactions from colloidal dispersions and characterized by ICP, elemental analysis, XRD, SEM, FT-IR, and solid-state NMR. Finally, cell viability and cytotoxicity tests are carried out in contact with MG-63 cells, which are human bone osteosarcoma cells.

The synthesis process and the results are interesting and well described. The characterizations provide a good description of the chemical structure of this innovative material.

In my opinion, this article could be worthy of publication in the journal Nanomaterials – MDPI after considering the following minor points:

 - The acronyms must be defined the first time they appear in the text (e.g., NMR, SEM…).

 - "Powder XRD" or "PXRD" is better than "XRPD", and more usual for the readership.

 - In line 242, please add the word "diffraction" to describe "X-ray diffraction analysis".

 - Energy dispersive X-ray spectroscopy is mentioned as EDS or EDX in the text. Please homogenize it, using only one acronym.

Thanks to the Reviewer for the suggestions. The text was modified accordingly.

   - more comparison with the literature on similar topics is necessary. Why are the properties of this innovative material valuable and more efficient than the others? For which applications?

We thank the Reviewer for the suggestion.

The text was modified accordingly as follows at page 2 line 66-75: The topotactic exchange method can be considered an alternative to the preparation, via intercalation reaction, of drug delivery systems based on ZP compounds. In this scenario, the intercalation of drugs as doxorubicin11 and 5-fluorouracil,12 clorexidine,13 methylene blue as photosensitizer14 and NorA efflux pump inhibitors15 was successfully performed on ZP supports. Thus ZPs, have shown promising as nanocarriers for drug delivery, and other biotechnological applications.1 This research work highlights the potential of the topotactic exchange method to prepare modified ZP in which ALN is covalently anchored on the surface of the supports. In this way it is possible to obtain an efficient load of the drug and a mediated release reducing at the same time the risk of the out-of-control release of BPs. 

For example, there is no discussion (and no references) in section 3.1, but only a description of some results. Why are these biological results interesting for the study? Please clarify it

The aim of this analysis was to assess the eligibility of these novel inorganic-organic nanoparticles in the bone tissue regeneration. The determination of cytotoxicity is one of the most pressing questions when engineered nanoparticles are employed in various fields including drug delivery, biosensors, cancer treatment and diagnostic tools. Therefore, the first step toward the effective use of new nanomaterials in biomedical field is the study on their potential toxicity.